# A New Insight into the Weight Gain Method to Monitor and Evaluate Lipid Peroxidation

**DOI:** 10.3390/foods14040700

**Published:** 2025-02-18

**Authors:** Haniye Pashaei, Reza Farhoosh

**Affiliations:** Department of Food Science and Technology, Faculty of Agriculture, Ferdowsi University of Mashhad, Mashhad P.O. Box 917751163, Iran

**Keywords:** kinetics, lipid hydroperoxides, peroxidation, peroxide value, weight gain

## Abstract

The kinetics of change in peroxide value and weight gain were simultaneously studied during the peroxidation of three vegetable oils of various chemical compositions. The initiation and propagation oxidizability parameters *O*_i_ (kg mEq^−1^ h^2^) and *r*_n_ (h^−1^) of the canola, corn, and peanut oils were calculated to be 412 and 0.0894, 160 and 0.0641, and 36.1 and 0.0304, respectively. The weight-based parameter *O*_iw_ ranked the oils (103,000, 51,000, and 8000% h^2^, respectively) as the parameter *O*_i_ did. However, the equivalent parameter *r*_nw_ (0.0680, 0.1198, and 0.1875 h^−1^, respectively) provided a reverse order than that of the parameter *r*_n_. This was attributed to the fact that the value of *r*_n_ represents the formation and decomposition of lipid hydroperoxides in general, whereas the value of *r*_nw_ actually represents the overall tendency of an oil to the formation and polymerization of lipid hydroperoxides.

## 1. Introduction

Peroxide value (PV) is the most well-known and frequently used oxidative indicator to analyze peroxidation in oils or fats and lipid-containing foods. It represents the concentration of lipid hydroperoxides (LOOH) as the primary oxidation products with basically no negative effects on the sensory quality and health of lipid materials. However, LOOH progressively increase over time and reach a critical concentration at which they decompose significantly into a wide variety of secondary oxidation products, making the system rancid and thereby nutritionally and sensorily unacceptable [1,2,3].

From a kinetic point of view, PV increases first linearly during an initiation phase, known as induction period (IP), and approaches the value PV_IP_ at the end of the phase. During a subsequent propagation phase, the value rises more rapidly and asymptotically reaches the maximum amount PV_max_. A sigmoidal model has recently been developed to describe the phase kinetically, providing the pseudo-first order composite rate constant *k*_c_ and the pseudo-second order decomposition rate constant *k*_d_ [4]. A usual peroxidation pattern such as this, with the two successive initiation and propagation phases, may occur stepwise, in which each step comprises the two specified phases [5]. In the end, PV decreases essentially as a result of the breakdown of LOOH at a higher rate than that of their formation [6]. This final stage, known as termination phase, possesses no practical importance and is rarely used for kinetic analysis in applied research [7].

Lipid peroxidation is necessarily accompanied by the uptake of oxygen, resulting in the oxidizing lipid gaining weight. Periodic measurement of weight gain is among the oldest methods for evaluating the extent of peroxidation [8]. Only simple equipment is used to provide a number of weight-based measures of oxidative stability. For example, the length of IP for oxygen uptake has been employed by plotting weight gain versus storage time. Occasionally, the time required to attain 0.5% weight increase has been taken as a measure of oxidative stability [8,9,10]. The magnitude of the slope at the steeply rising part of the curve has also been used to indicate the speed of the degradation of oxidizing substrates [11]. In some cases, the maximum increase in the weight of the sample has been considered to explain a higher probability of the oxidative phenomena occurring [11,12]. Therefore, this physical technique of low instrumentation cost, offering a high capacity and processing speed of samples with no limitation [10], has been suggested to be extended to more sophisticated continuous monitoring of mass and energy changes, as in thermogravimetry (TG) and/or differential scanning calorimetry (DSC) [8].

A literature review showed no comprehensive approach to reliably analyze changes in weight during lipid peroxidation. However, the curves of weight gain often indicate successive linear and sigmoidal stages over time [8,9,10,11,12]. Accordingly, mathematical functions of these kinds would provide us with some quantitative information on how lipid peroxidation proceeds [13]. In addition, no scientific report was found on the coincidence of the kinetics of change in weight and PV during different phases of lipid peroxidation. Hence, the present study aimed to reveal whether the quantitative measures characterizing weight gain would be able to provide results relative to those from the kinetics of change in PV. To do this, the kinetics of change in weight and PV were simultaneously studied during the peroxidation of some vegetable oils (canola, corn, and peanut) of various chemical compositions.

## 2. Materials and Methods

### 2.1. Materials

Commercial canola and corn oils and cold-pressed peanut oil were purchased from local manufacturing units and stored at −18 °C until analysis. All chemicals, reagents, and solvents used in the study were of analytical grade and purchased from Merck (Darmstadt, Germany) and Sigma-Aldrich (St. Louis, MO, USA).

### 2.2. Fatty Acid Composition

The solutions of the oils in hexane (0.3 g in 7 mL) were shaken with 7 mL of 2 M methanolic KOH at 50–55 °C for 15 min. The upper layer was mixed with anhydrous Na_2_SO_4_ after 5 min and was then filtered. The solutions were injected into an HP-5890 chromatograph (Hewlett-Packard, Palo Alto, CA, USA). A BPX 70 capillary column (60 m × 0.22 mm I.D., 0.2 mm film thickness) and an FID detector were used for separation. The flow rate of He as a carrier gas was 0.7 mL min^−1^. The oven, injector, and detector were set at 198 °C, 280 °C, and 250 °C, respectively. To judge the overall tendency of the fatty acid compositions to undergo peroxidation, the calculated oxidizability (Cox) value as [C18:1 + 10.3 (C18:2) + 21.6 (C18:3)]/100 [14] and the ratio between the polyunsaturated (PUFA) and saturated (SFA) fatty acids, known as the polyene index, were calculated [15].

### 2.3. Initial Quality Indicators of the Oils

#### 2.3.1. Peroxide Value (PV)

PV (mEq of O_2_ per kg of oil) was measured spectrophotometrically using a UV-VIS instrument (model 160A Shimadzu, Kyoto, Japan). The oils (0.01–0.3 g) were dissolved in 9.8 mL of chloroform–methanol (7:3 *v*/*v*) on a vortex mixer (2–4 s). Next, ammonium thiocyanate solution (50 µL, 30% *w*/*v*) was added and vortexed again (2–4 s). Then, 50 µL of iron (II) chloride solution was added, and after vortexing (2–4 s) and then incubation at room temperature (5 min), absorbance was read at 500 nm [16].

#### 2.3.2. Conjugated Diene Value (CDV)

The oils (0.01–0.03 g) were dissolved in 10 mL of *n*-hexane (HPLC grade) and their absorbance were read at 234 nm [17]. An extinction coefficient of 29,000 L mol^−1^ cm^−1^ was used to calculate CDV (μmol g^−1^) [18].

#### 2.3.3. Carbonyl Value (CV)

The oils (0.04–1.00 g) were dissolved in 10 mL of purified 2-propanol containing 0.4 mg mL^−1^ of triphenylphosphine. The solutions (1 mL) were mixed with 2,4-DNPH solutions (1 mL: 50 mg in 100 mL of 2-propanol containing 3.5 mL of 370 g L^−1^ HCl) in 15 mL test tubes. The test tubes were capped and heated for 20 min at 40 °C. After cooling in a water bath, 8 mL of KOH solutions (20 g L^−1^) were added. The solutions were centrifuged (2000× *g* for 5 min) at room temperature. Absorbance of the upper layers was read at 420 nm against the purified 2-propanol as a blank. 2,4-Decadienal in the purified 2-propanol (50–500 μmol L^−1^) was used to draw the calibration curve [19].

#### 2.3.4. Acid Value (AV)

AV (mg of KOH to neutralize free fatty acids in one g of oil) was determined by dissolving 10 g of the oils in 50 mL of chloroform–ethanol (50:50 *v*/*v*) and titrating with 0.1 M ethanolic KOH [20].

### 2.4. Total Tocopherols Content

The oils (100 mg) plus 5 mL of toluene were poured into a 10 mL volumetric flask. Then, 3.5 mL of 2,2′-bipyridine (0.07% *w*/*v* in 95% aqueous ethanol) and 0.5 mL of FeCl_3_.6H_2_O (0.2% *w*/*v* in 95% aqueous ethanol) were added in sequence. The volume of the solution was made up to 10 mL with 95% aqueous ethanol. The absorbance was read at 520 nm after 1 min against a blank prepared as above with no oil. Alpha-tocopherol in toluene (0–240 μg mL^−1^) was used to draw the calibration curve [21].

### 2.5. Total Phenolics Content

The oils (2.5 g) were dissolved in 2.5 mL of *n*-hexane and extracted three times (3 × 2.5 mL) with CH_3_OH:H_2_O (80:20 *v*/*v*) in a centrifuge at 5000 rpm for 5 min. Then, 2.5 mL of Folin–Ciocalteau reagent and 5 mL of 7.5% Na_2_CO_3_ were added to the extract in a 50 mL volumetric flask, reaching the final volume with distilled water. The solutions were maintained for 2 h in the dark and the absorbance was read at 765 nm. Gallic acid in methanol (0.04–0.40 mg mL^−1^) was used to draw the calibration curve [22].

### 2.6. Peroxidation

The oils were stored in an oxygen pressure-independent oxidative regime [23] in a series of accurately weighed open Petri dishes (~7 cm in diameter) containing a 1 mm layer of about 3.6 g of oil in an oven (A & D Co., Ann Arbor, MI, USA) at 90 ± 3 °C. At specified time intervals, one Petri dish was removed from the oven, allowed to cool at room temperature for 10 min in a desiccator, reweighed by an analytical balance measuring to 0.1 milligrams [10], and then the PV of the oil was measured [16].

### 2.7. Kinetic Data Analyses

#### 2.7.1. PV

Changes in PV (mEq kg^−1^) were plotted over time *t* (h). The linear Equation (1) was fitted to the initiation peroxidation phase (Figure 1 and Figure 2):PV = *k*_IP_ *t* + PV_0_(1)
where *k*_IP_ (mEq kg^−1^ h^−1^) and PV_0_ (mEq kg^−1^) are a pseudo-zero order rate constant and PV at *t* = 0, respectively [24]. The linear Equation (2) was obtained by the coordinates of the turning point and slope at the point of a third-power function (PV = *at*^3^ + *bt*^2^ + *ct* + *d*) fitted to the second initiation peroxidation phase (Figure 1):(2)$$\;\mathrm{PV}=\left(-\frac{b^{2}}{3a}+c\right)t-\frac{b^{3}}{27a^{2}}+d$$
where *k*_IP_ = (−*b*^2^/3*a*) + *c* and PV_0_ = (−*b*^3^/27*a*^2^) + *d* [5]. The sigmoidal Equation (3) was fitted to the propagation peroxidation phases:(3)$$\mathrm{PV}=\frac{k_{c}}{\exp\left[k_{c}\;\left(C-t\right)\right]+k_{d}}$$
where *k_c_* (h^−1^) and *k_d_* (kg mEq^−1^ h^−1^) are pseudo-first and pseudo-second order rate constants, respectively. C is an integration constant. The maximum level of PV (PV_max_, mEq kg^−1^) was calculated from the ratio *k_c_*/*k_d_*. The time attaining PV_max_ was *t*_max_ = (2 + *k_c_*C − ln*k_d_*)/*k_c_*. The maximum rate of change in PV (*r*_max_, mEq kg^−1^ h^−1^) and its normalized form (*r*_n_, h^−1^), representing the propagation oxidizability parameter, were computed using the ratios *k_c_*^2^/4*k_d_* and *r*_max_/PV_max_, respectively [24]. Induction period (IP, h) and PV at IP (PV_IP_, mEq kg^−1^) were provided as follows:(4)$$\mathrm{IP}=\frac{{\mathrm{PV}}_{\max}-\left(r_{\max}\times t_{\max}\right)-{\mathrm{PV}}_{0}}{k_{\mathrm{IP}}-r_{\max}}$$PV_IP_ = *k*_IP_ IP + PV_0_(5)

The initiation oxidizability parameter *O*_i_ (kg mEq^−1^ h^2^) and duration of the propagation phase *t*_p_ (h) were given by the ratio IP/*k*_IP_ and *t*_max_ − IP, respectively [4].

#### 2.7.2. Weight Gain

Changes in weight (W, %) were plotted over time *t* (h). The linear Equation (6) was fitted to the initial linear stage:(6)$$\;W=r_{\mathrm{IP}}\;t+W_{0}$$
where *r*_IP_ (% h^−1^) and W_0_ (%) are the zero-order rate constant of change in W and W at *t* = 0, respectively (Figure 1 and Figure 2). Similarly, a third-power function (W = *at*^3^ + *bt*^2^ + *ct* + *d*) and then the linear Equation (2) were used to calculate the second values of *r*_IP_ and W_0_ for the canola oil of a two-step kinetic W curve (Figure 2). The sigmoidal Equation (7) was fitted to the kinetic data points of W curve at its steeply rising part:(7)$$\;W=a+\frac{b}{1+e^{\left(\frac{c-t}{d}\right)}}$$
where *a*, *b*, *c*, and *d* are the equation parameters. The finite value W_max_ (%), maximum rate of change in W (*r*_maxw_, % h^−1^), and normalized *r*_maxw_ (*r*_nw_, h^−1^) were calculated from *a* + *b*, 0.25 *b*/*d*, and *b*/4*d*(*a* + *b*), respectively. The time at which W reaches W_max_ (*t*_maxw_, h) was obtained from *c* + 2*d*. The duration of the linear stage (IP_W_, h) was given by Equation (8):(8)$${\mathrm{IP}}_{W}=\frac{0.25bc+d\left(W_{0}-a-0.5b\right)}{0.25b-\left(d\times r_{\mathrm{IP}}\right)}$$

The parameter *O*_iw_ (% h^2^) and duration of the sigmoidal stage *t*_s_ (h) were given by the ratio IP_w_/*r*_IP_ and *t*_maxw_ − IP_w_, respectively [24].

### 2.8. Statistical Analysis

The measurements were carried out in triplicate. The data underwent an analysis of variance (ANOVA). ANOVA was performed using MStatC version 1.1.0 and SlideWrite version 7.0. Duncan’s multiple-range tests were used to determine significant differences in the means at *p <* 0.05.

## 3. Results and Discussion

### 3.1. Chemical Composition of the Oils

Table 1 shows the major fatty acids determined in the oils experimented with, which were in accordance with those reported in the literature for canola [25], corn [26], and peanut [27] oils. As expected, the lowest total amount of SFA (palmitic, C16:0, and stearic, C18:0) was found in the canola oil. The relatively noticeable quantities of the monounsaturated fatty acid (MUFA) oleic (C18:1∆^9^) were observed in both the canola and the peanut oils. The corn oil instead contained a very high level of the PUFA linoleic (C18:2∆^9,12^). And by far the largest extent of the PUFA linolenic (C18:3∆^9,12,3^) was measured in the canola oil. The relative rate of oxidation for stearic, oleic, linoleic, and linolenic acids has been reported to be 1:100:1200:2500 [28]. Thus, the oxidative stability of the fatty acid compositions is definitely affected by the proportion of each fatty acid. The fatty acid composition of peanut oil, with the Cox value and polyene index of 2.82 and 1.41, respectively, is naturally expected to have the highest oxidative stability, followed by those of the canola (4.47 and 3.60, respectively) and corn (5.96 and 3.74, respectively) oils.

*Trans* fatty acids, regardless of some exceptions, are the unnatural constituents in edible fats and oils, resulting from the deteriorative chemical reactions which may occur during storage and many technological operations, such as refining, partial hydrogenation, or frying [29]. There is no doubt that the extent and diversity of *trans* fatty acids in an oil essentially depend upon the quality and intensity of the operations used as well as the inherent oxidative stability of the oil, coming from many factors including fatty acid composition. As shown in Table 1, a higher total amount of *trans* fatty acids was measured in the corn than in the canola oil, and the peanut oil almost lacked any significant amount of *trans* fatty acids. The peanut oil was a cold-pressed extracted oil which had naturally undergone the least technological operations and also possessed a fatty acid composition of relatively higher oxidative stability than those of the other two oils.

In addition to fatty acid composition, the initial quality indicators PV, CDV, CV, and AV can also affect the oxidative stability of an oil. The PV and AV of the oils were roughly in the ranges promulgated by the CODEX standard (below 10 mEq kg^−1^ and 0.6 mg g^−1^, respectively) [30]. The peanut oil had the highest initial PV but interestingly the lowest initial CDV among the vegetable oils studied (Table 1). This was in line with the least total amount of *trans* fatty acids in the peanut oil and then in the canola and corn oils, respectively. Hence, the apparently lower PVs in the canola and corn oils might have been due to the decomposition of their preformed LOOH into the conjugated diene non-hydroperoxides, as affected by the commercial processing activities [31]. This necessitates the need to consider initial CDV as a quality indicator, capable of showing the potential impact of technological operations. The CV of the oils was below the maximum levels suggested for well-refined oils (0.5–2.0 µmol g^−1^) [32].

Tocopherols and phenolic compounds as potential natural antioxidants may exert the same contribution as the other factors mentioned above, or even higher, in stabilizing edible fats and oils. Their total content was generally higher in the canola than in the corn and peanut oils, respectively (Table 1). Moreover, it should be noticed that the canola and corn oils contained some additives, including the synthetic antioxidant TBHQ as well as the synergist citric acid.

### 3.2. Kinetics of Change in PV

The canola oil exhibited a two-step peroxidation prior to attaining the termination phase (Figure 1A) whereas the corn and peanut oils showed usual multiphase peroxidations, including the initiation, propagation, and termination phases (Figure 2A,B). This was in agreement with the patterns observed in our recent studies, indicating that stepwise peroxidation is more likely to occur when the oxidizing system is of low homogeneity in its fatty acid composition [5]. Canola oil, with a relatively high percentage of oleic acid and a notably large amount of linolenic acid (Table 1), which is oxidized 25 times faster than oleic acid [28], was highly prone to undergo stepwise peroxidation. The kinetic parameters characterizing the two separate steps of oil peroxidation are presented in Table 2. However, it has been shown that only the first peroxidation step is crucial from sensory and toxicological points of view, and afterward, the oil must be discarded [5]. Thus, the kinetic parameters representing the second peroxidation step, which may be worth considering kinetically (see [5]), have not been taken into account in the present study.

Table 2 shows the kinetic parameters of the whole practical range of the peroxidation of the oils at 90 °C. The initiation oxidizabiliy parameter *O*_i_, unifying the classic parameter IP and the rate constant *k*_IP_, indicates that the canola oil (412 kg mEq^−1^ h^2^) displayed significantly higher resistance to the generation of peroxyl radicals during the phase [33]. The corn and peanut oils yielded significantly lower *O*_i_ values (160 and 36.1 kg mEq^−1^ h^2^, respectively). This can simply be explained by considering the chemical composition data shown in Table 1. The canola oil possessed a more oxidatively stable fatty acid composition as well as a significantly higher total phenolics content compared with the corn oil. The peanut oil, however, which had by far the lowest Cox value, contained the lowest total content of tocopherols and phenolic compounds and lacked any added synthetic antioxidant or synergist.

The canola, corn, and peanut oils passed through PVs of 5.21, 4.72, and 8.28 mEq kg^−1^, respectively, when transitioning from the initiation to the propagation phase (PV_IP_, Table 2). These were much lower than that (~20 mEq kg^−1^) suggested by Crapiste et al. [34] as a safety threshold value at the end of IP. The highest duration of the propagation phase (*t*_p_), which is the time range from IP to *t*_max_, was 8.79 h for the canola oil, followed by 14.3 h and 28.2 h for the corn and peanut oils, respectively (Table 2). This was in reverse of the decreasing trend observed for the corresponding values of IP (from 36.2 h to 13.1 h). The value of *t*_p_ is normally much smaller than that of IP unless the lipid system is too oxidizable and/or exposed to very harsh oxidative conditions. It is kinetically related to the rate of peroxidation in the phase (*r*_max_) as well as PV_max_, which is in turn a function of the ratio *k_c_*/*k_d_* [24]. The value of *r*_n_, which represents the oxidizability of lipid systems during the propagation phase, demonstrated higher oxidative stability for the peanut oil (304 × 10^−4^ h^−1^) than for the corn (641 × 10^−4^ h^−1^) and canola (894 × 10^−4^ h^−1^) oils, respectively. With respect to the fact that antioxidants are mostly consumed during IP [33], the order of the observed propagation oxidizabilities can be explained mainly in terms of the fatty acid composition and the initial quality indicators as well. The value of *r*_n_ for the peanut oil being significantly lower than all others can be attributed to the oil having by far the lowest Cox value and polyene index. The canola and corn oils were of statistically the same values of polyene index, which concomitantly takes into account the contributions of PUFA and SFA. However, the corn oil contained the bigger amount of the synergist citric acid by a factor of 2 as well as having significantly lower PV and AV. The latter two initial quality indicators may possess remarkable contributions to LOOH production.

### 3.3. Kinetics of Change in Weight

Figure 1B,C and Figure 2C,D show that the weight of the oils decreased to some extent at early stages and then increased continuously with peroxidation. The initial weight losses have been attributed to the partial evaporation of humidity and the organic compounds of high volatility [34]. The same kinetic patterns of change in PV were observed in weight as well, namely the stepwise and usual multiphase variations for the canola oil and the corn and peanut oils, respectively. Table 3 presents the kinetic parameters characterizing the change in weight of the oils over the process.

As shown in Figure 1B and also in its extended form for the smaller percentages of weight gain (Figure 1C), after the IP_W_ of 48.6 h, the canola oil entered into the first sigmoidal stage (*t*_s_ = 15.8 h) and attained the end at the *t*_maxw_ of 64.4 h (Table 3). This means that the oil weight increased almost slightly and then rapidly during the first (from IP = 36.2 h to *t*_max_ = 45 h) and second (from IP = 48.6 h to *t*_max_ = 56.4 h) propagation phases of peroxidation, respectively (Table 2). Most importantly, during the termination phase of peroxidation (after *t*_max_ = 56.4 h), where the curve is plateaued (Figure 1A), the oil weight still increased and reached the second sigmoidal stage (Figure 1B,C) with a much higher rate (*r*_maxw_ = 0.1861% h^−1^) than before (*r*_maxw_ = 0.0191% h^−1^) (Table 3). Such a periodic unconformity between the kinetic curves of change in weight and PV was also observed for the corn and peanut oils (Figure 2). This reveals that the kinetics of change in weight are not consistently the same in PV (see below).

The value of IP_W_ for each of the oils (Table 3) was notably bigger than the equivalent values of IP (Table 2). As illustrated in Figure 1 and Figure 2, both the initiation and propagation phases of lipid peroxidation coincided with the initial linear stages of the kinetic curves of change in weight. Approximately the same values of *t*_max_ (Table 2) and IP_W_ (Table 3) for each oil alone affirms such a phenomenon. This must essentially be due to the more remarkable decomposition of LOOH in the propagation than in the initiation peroxidation phase into volatile products (e.g., carbon dioxide, short chain acids, ketones, aldehydes, and alcohols) which easily diffuse out and prevent gaining weight steeply [12]. However, in contrast to IP, the values of IP_W_ provided no significant difference between the corn and peanut oils. As shown in Figure 2 and Table 2, peroxidation progressed more slowly in the propagation phase of the peanut oil (*r*_max_ = 1.79 mEq kg^−1^ h^−1^) than in that of the corn oil (*r*_max_ = 3.43 mEq kg^−1^ h^−1^), leading to lesser amount of oxygen uptake in the former. In other words, the value of *t*_p_ (14.3 h and 28.2 h for the corn and peanut oils, respectively) may remarkably affect the length of IP_W_. This was in full accordance with the much lower degree of polyunsaturation in the peanut oil (Table 1). Matikainen et al. [11] demonstrated a direct quantitative relationship between oxygen absorption and the number of reactive allylic positions per molecule, such that, on average, four and three oxygen atoms are joined to one molecule of methyl linolenate and methyl linoleate, respectively. Interestingly, the parameter *O*_iw_, which unifies the values of IP_W_ and *r*_IP_, significantly ranked the oils as the initiation oxidizability parameter *O*_i_ did.

The sigmoidal stages of gaining weight (after IP_W_, Table 3), which coincided with the termination peroxidation phases (after *t*_max_, Table 2), indicates that the rapid uptake of oxygen in the kinetic curves of weight gain implies dramatic decomposition of LOOH and, as a result, deteriorations in the sensory and nutritional attributes of an oil that are too high. The rates of the curves in the sigmoidal stage (*r*_maxw_, Table 3) reveal that the speed of oxygen uptake in the peanut oil was noticeably higher than those in the canola and corn oils, with no significant difference. The peanut oil also displayed a significantly bigger value of W_max_ compared with the other two oils. Naturally, fatty acids of a higher unsaturation degree generate more unstable LOOH that decompose more easily into the products of higher volatility and are less likely to undergo polymerization through intermolecular addition or cross-linking reactions [11,12]. Considering the smaller Cox value and polyene index in the peanut, canola, and corn oils, respectively (Table 1), the normalized value of *r*_maxw_ (*r*_nw_ = *r*_maxw_/W_max_, Table 3) effectively demonstrated the higher potency of the oils to undergo polymerization in the same order.

## 4. Conclusions

The present study indicated that the kinetics of change in weight and PV do not follow exactly the same patterns during lipid peroxidation. However, the weight gain method can provide some complementary information to PV regarding the formation of LOOH and their transformation to volatiles and polymeric products. Similarly to the initiation oxidizability parameter *O*_i_, the kinetic parameter *O*_iw_, which stands for the linear stage of the kinetic curves of weight gain, can be used to compare the oxidative stability of lipid systems. Furthermore, unlike the propagation oxidizability parameter *r*_n_, which represents LOOH formation and decomposition overall, the parameter *r*_nw_ can be used as a measure of overall tendency to LOOH formation and polymerization. Undoubtedly, further study on a wider range of edible fats and oils under various antioxidative and pro-oxidative conditions would provide us with more certainty on the practical applications of the weight-based kinetic parameters, which can be obtained very simply and inexpensively.

## Figures and Tables

**Figure 1 foods-14-00700-f001:**
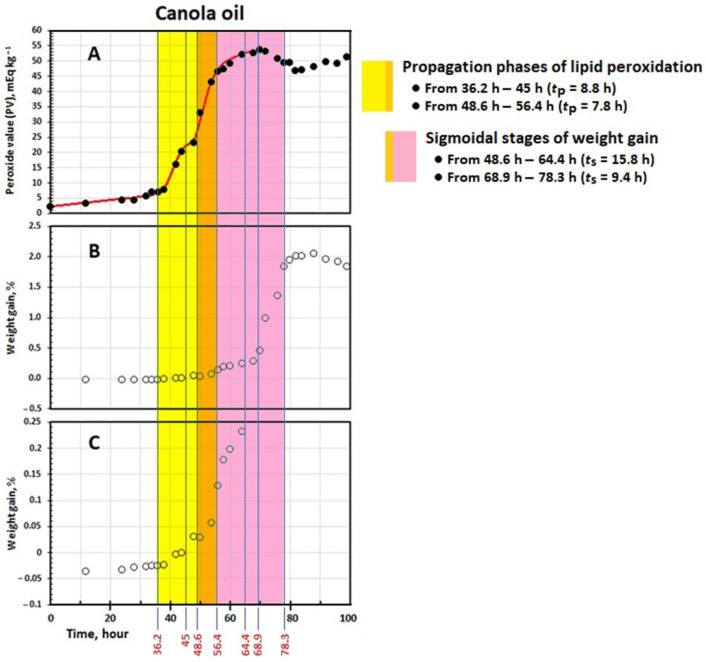
Kinetic curves of change in peroxide value (PV, (**A**)) and weight (**B**,**C**) during peroxidation of the canola oil at 90 °C.

**Figure 2 foods-14-00700-f002:**
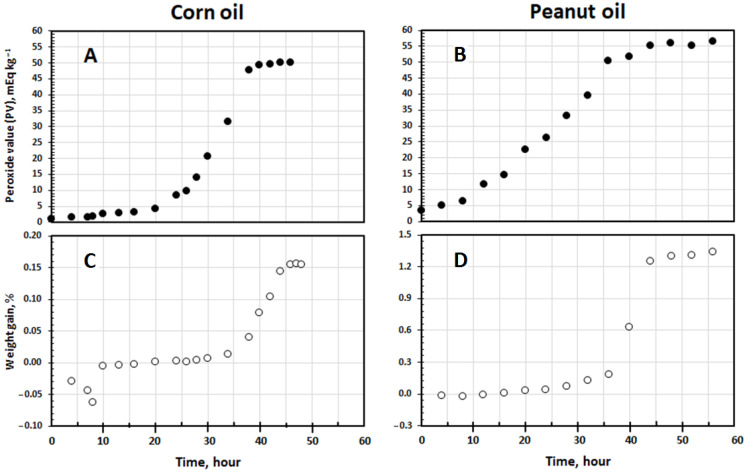
Kinetic curves of change in peroxide value (PV; (**A**,**B**)) and weight (**C**,**D**) during peroxidation of the corn and peanut oils at 90 °C.

**Table 1 foods-14-00700-t001:** The chemical composition data and initial quality indicators of the canola, corn, and peanut oils.

	Oil Sample		
	Canola	Corn	Peanut
Major fatty acids (%*w*/*v*)			
C16:0	4.90 ± 0.1 ^c^	11.0 ± 0.2 ^a^	9.64 ± 0.08 ^b^
C18:0	2.03 ± 0.01 ^c^	2.40 ± 0.01 ^b^	3.65 ± 0.00 ^a^
C18:1 *cis*-∆^9^	61.1 ± 0.1 ^b^	31.3 ± 0.0 ^c^	62.6 ± 0.2 ^a^
C18:1 *trans*	0.11 ± 0.00 ^a^	0.03 ± 0.01 ^b^	-
C18:2 *cis*-∆^9,12^	20.9 ± 0.0 ^b^	52.3 ± 0.2 ^a^	21.0 ± 0.1 ^b^
C18:2 *trans*	0.10 ± 0.00 ^b^	0.65 ± 0.04 ^a^	0.02 ± 0.00 ^c^
C18:3 *cis*-∆^9,12,15^	7.42 ± 0.02 ^a^	0.81 ± 0.00 ^b^	0.12 ± 0.01 ^c^
C18:3 *trans*	0.20 ± 0.00 ^a^	0.08 ± 0.01 ^b^	-
Calculated oxidizability (Cox) value	4.47 ± 0.00 ^b^	5.96 ± 0.02 ^a^	2.82 ± 0.01 ^c^
Polyene index	3.60 ± 0.03 ^a^	3.74 ± 0.06 ^a^	1.41 ± 0.00 ^b^
Peroxide value (PV, mEq kg^−1^)	2.03 ± 0.05 ^b^	0.96 ± 0.04 ^c^	3.52 ± 0.11 ^a^
Conjugated diene value (CDV, µmol g^−1^)	13.0 ± 0.5 ^b^	14.5 ± 0.2 ^a^	6.70 ± 0.08 ^c^
Carbonyl value (CV, µmol g^−1^)	0.28 ± 0.04 ^b^	0.74 ± 0.05 ^a^	0.32 ± 0.05 ^b^
Acid value (AV, mg g^−1^)	0.47 ± 0.09 ^b^	0.25 ± 0.04 ^c^	0.78 ± 0.04 ^a^
Total tocopherols content (mg kg^−1^)	600 ± 12 ^a^	592 ± 8 ^a^	210 ± 5 ^b^
Total phenolics content (mg kg^−1^)	261 ± 1 ^a^	205 ± 2 ^b^	57 ± 2 ^c^
Additives ^†^			
*tert*-Butylhydroquinone (TBHQ, mg kg^−1^)	75	75	-
Citric acid (mg kg^−1^)	50	100	-

Means ± SD (standard deviation) within a row with the same lowercase letters are not significantly different at *p* < 0.05. ^†^ Taken from the labels of the oil bottles.

**Table 2 foods-14-00700-t002:** Kinetics of change in peroxide value (PV, mEq kg^−1^) during peroxidation of the canola, corn, and peanut oils at 90 °C.

	Canola Oil		Corn Oil	Peanut Oil
	Step 1	Step 2		
*Initiation phase*				
IP (h)	36.2 ± 0.1 ^b^	48.6 ± 0.3 ^a^	25.4 ± 0.1 ^c^	13.1 ± 0.4 ^d^
*k*_IP_ (mEq kg^−1^ h^−1^)	0.09 ± 0.01 ^d^	1.34 ± 0.40 ^a^	0.16 ± 0.01 ^c^	0.36 ± 0.03 ^b^
*O*_i_ (kg mEq^−1^ h^2^)	412 ± 44 ^a^	38.8 ± 1.3 ^c^	160 ± 8 ^b^	36.1 ± 3.1 ^c^
PV_IP_ (mEq kg^−1^)	5.21 ± 0.32 ^c^	26.4 ± 2.2 ^a^	4.72 ± 0.18 ^c^	8.28 ± 0.37 ^b^
*Propagation phase*				
*t*_p_ (h)	8.79 ± 0.40 ^c^	7.85 ± 0.85 ^c^	14.3 ± 0.8 ^b^	28.2 ± 0.2 ^a^
*t*_max_ (h)	45.0 ± 0.4 ^b^	56.4 ± 0.6 ^a^	39.6 ± 0.8 ^c^	41.4 ± 0.4 ^c^
*k_c_* (h^−1^)	0.36 ± 0.02 ^a^	0.25 ± 0.02 ^b^	0.26 ± 0.02 ^b^	0.12 ± 0.00 ^c^
*k_d_* (kg mEq^−1^ h^−1^)	147 ± 6 (×10^−4^) ^a^	48.3 ± 4.1 (×10^−4^) ^b^	48.0 ± 3.4 (×10^−4^) ^b^	20.7 ± 0.5 (×10^−4^) ^c^
PV_max_ (mEq kg^−1^)	24.3 ± 0.7 ^c^	52.8 ± 1.7 ^b^	53.5 ± 0.8 ^b^	58.8 ± 1.1 ^a^
*r*_max_ (mEq kg^−1^ h^−1^)	2.17 ± 0.11 ^b^	3.35 ± 0.12 ^a^	3.43 ± 0.15 ^a^	1.79 ± 0.03 ^c^
*r*_n_ (h^−1^)	894 ± 34 (×10^−4^) ^a^	636 ± 37 (×10^−4^) ^b^	641 ± 36 (×10−4) ^b^	304 ± 3 (×10^−4^) ^c^

Means ± SD (standard deviation) within a row with the same lowercase letters are not significantly different at *p* < 0.05. IP—induction period; *k*_IP_—pseudo-zero order rate constant of initiation phase; *O*_i_—initiation oxidizability parameter; PV_IP_—PV at IP; *t*_p_—duration of propagation phase; *t*_max_—the time at which PV reaches its maximum value (PV_max_); *k_c_*—composite (pseudo-first order) rate constant; *k_d_*—decomposition (pseudo-second order) rate constant; PV_max_—maximum level of PV; *r*_max_—maximum rate of change in PV; *r*_n_—normalized *r*_max_, propagation oxidizability parameter.

**Table 3 foods-14-00700-t003:** Kinetics of change in weight (W, %) during peroxidation of the canola, corn, and peanut oils at 90 °C.

	Canola Oil		Corn Oil	Peanut Oil
	Step 1	Step 2		
IP_w_ (h)	48.6 ± 0.1 ^b^	68.9 ± 0.2 ^a^	36.8 ± 0.9 ^c^	37.7 ± 0.8 ^c^
*t*_s_ (h)	15.8 ± 1.1 ^a^	9.37 ± 0.35 ^b^	8.60 ± 0.36 ^b^	5.25 ± 0.83 ^c^
*t*_maxw_ (h)	64.4 ± 1.1 ^b^	78.3 ± 0.5 ^a^	45.4 ± 2.6 ^c^	42.9 ± 0.1 ^c^
W_max_ (%)	0.28 ± 0.01 ^c^	1.98 ± 0.06 ^a^	0.17 ± 0.02 ^d^	1.31 ± 0.01 ^b^
*r*_IP_ (% h^−1^)	475 ± 55 (×10^−6^) ^d^	5836 ± 263 (×10^−6^) ^a^	724 ± 30 (×10^−6^) ^c^	4716 ± 123 (×10^−6^) ^b^
*r*_maxw_ (% h^−1^)	0.0191 ± 0.0005 ^c^	0.1861 ± 0.0054 ^b^	0.0202 ± 0.0058 ^c^	0.2456 ± 0.0098 ^a^
*O*_iw_ (% h^2^)	10.3 ± 1.3 (×10^4^) ^a^	1.2 ± 0.2 (×10^4^) ^c^	5.1 ± 0.3 (×10^4^) ^b^	0.8 ± 0.0 (×10^4^) ^d^
*r*_nw_ (h^−1^)	0.0680 ± 0.0052 ^d^	0.0940 ± 0.0050 ^c^	0.1198 ± 0.0461 ^b^	0.1875 ± 0.0543 ^a^

Means ± SD (standard deviation) within a row with the same lowercase letters are not significantly different at *p* < 0.05. IP_w_—induction period; *t*_s_—duration of the sigmoidal stage; *t*_maxw_—the time at which W reaches its maximum value (W_max_); *r*_IP_—rate of change in W at IP_w_; *r*_maxw_—maximum rate of change in W; *O*_iw_—the ratio between IP_W_ and *r*_IP_; *r*_nw_—normalized *r*_maxw_.

## Data Availability

The original contributions presented in this study are included in the article. Further inquiries can be directed to the corresponding author.

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
