# Peer review of "A New Insight into the Weight Gain Method to Monitor and Evaluate Lipid Peroxidation"

_foods, 2025, doi:10.3390/foods14040700_

Round 1
Reviewer 1 Report
Comments and Suggestions for Authors
This article explores the kinetics of lipid peroxidation in vegetable oils, focusing on the relationship between weight gain and peroxide value (PV) changes. The study concludes that weight gain measurements can be a useful tool for evaluating oxidative stability, offering insights beyond those provided by PV alone.This study uniquely combines the measurement of weight gain with PV changes. By monitoring both parameters simultaneously, the researchers gain insights into the overall oxidative process, including the formation of volatile compounds and polymerization products that are not captured by PV alone.
1. While the study highlights the benefits of the weight gain method, it does not compare its effectiveness with other established methods for evaluating lipid peroxidation, such as Rancimat or differential scanning calorimetry (DSC).
2. Please fix the formatting problem in Table 2.
3. Standardize the format of the references to ensure compliance with the journal's requirements.
4. While the study presents a wealth of data, it lacks a deeper exploration of the chemical mechanisms underlying the observed kinetic patterns.
5. Ensure the format of significance marks for tabular data is uniform.
6. Briefly expound The innovation of this manuscript and its importance in the study of lipid oxidation.
7. Lines 225-230, referring to natural antioxidants or additives in cooking oils, consider setting up different antioxidant treatment groups or adding different concentrations of antioxidants to further study the effect of antioxidant factors on PV and weight change dynamics.
8. Can the weight change method replace the PV method? It is suggested to combine with other analytical methods, such as gas chromatography-mass spectrometry (GC-MS), to analyze lipid oxidation products more comprehensively and further verify the effectiveness of the weight change method.
Comments on the Quality of English Language
none
Author Response
The authors are grateful to the reviewer for his/her constructive evaluation of the manuscript entitled “A new insight into the weight gain method to monitor and evaluate lipid peroxidation” with the reference number foods-3430154.
Reviewer's comments and responses
This article explores the kinetics of lipid peroxidation in vegetable oils, focusing on the relationship between weight gain and peroxide value (PV) changes. The study concludes that weight gain measurements can be a useful tool for evaluating oxidative stability, offering insights beyond those provided by PV alone. This study uniquely combines the measurement of weight gain with PV changes. By monitoring both parameters simultaneously, the researchers gain insights into the overall oxidative process, including the formation of volatile compounds and polymerization products that are not captured by PV alone.
- While the study highlights the benefits of the weight gain method, it does not compare its effectiveness with other established methods for evaluating lipid peroxidation, such as Rancimat or differential scanning calorimetry (DSC).
Kindly, such a comparison has not been carried out to date, although it is a quite good idea and naturally requires doing a separate research study in the future.
- Please fix the formatting problem in Table 2.
It was fixed.
- Standardize the format of the references to ensure compliance with the journal's requirements.
The format was rechecked.
- While the study presents a wealth of data, it lacks a deeper exploration of the chemical mechanisms underlying the observed kinetic patterns.
We agree with the respected reviewer but it should be noticed that the present study provides the first kinetic findings regarding the change in weight during peroxidation as well as their practical application in parallel to the PV measurement. Definitely, elucidation of the chemical mechanisms underlying the observed kinetic patterns requires much more time and also some additional studies of this view in the future.
5.Ensure the format of significance marks for tabular data is uniform.
It was done.
6.Briefly expound the innovation of this manuscript and its importance in the study of lipid oxidation.
Kindly, we did our best to state about the innovation and practical importance of our findings in the Conclusion section. We hope this is satisfactory. Anyway, we are also ready to add any other points that the respected reviewer needs to be added.
7. Lines 225-230, referring to natural antioxidants or additives in cooking oils, consider setting up different antioxidant treatment groups or adding different concentrations of antioxidants to further study the effect of antioxidant factors on PV and weight change dynamics.
The oxidative stability of lipid systems is generally affected by their (1) fatty acid composition, (2) initial quality indicators (e.g. PV, AV, CD, and CV), and existing synthetic (e.g. BHA, BHT, and TBHQ) and/or natural antioxidants (e.g. tocopherols and phenolic compounds). As can be seen in different points of the manuscript, this compositional information helped us very well to interpret the oxidizabilities in terms of the calculated kinetic parameters.
8.Can the weight change method replace the PV method? It is suggested to combine with other analytical methods, such as gas chromatography-mass spectrometry (GC-MS), to analyze lipid oxidation products more comprehensively and further verify the effectiveness of the weight change method.
As stated in the sections R&D and Conclusions, the kinetic parameter Oiw (weight gain method) would be able to simulate well the kinetic parameter Oi (PV method), and the kinetic parameter rnw (for LOOH formation and polymerization) can provide some extra information in addition to those resulting from the parameter rn (for LOOH formation and decomposition). Anyway, as stated in the Conclusions section, study on a wider range of edible fats and oils under various antioxidative and pro-oxidative conditions would provide us with more certainty on the practical applications of the weight gain method. As for the respected reviewer’s suggestion, the combination of the weight gain method with the other analytical methods is a quite good idea that should be followed in future studies.
Reviewer 2 Report
Comments and Suggestions for Authors
This study tried to test a new method based on the weight gain of oil upon oxidation. However, the study focus a lot on the characterization of the chosen three oils, including fatty acid composition, PV values, CDV, CV, AV, Total Tocopherols Content, Total Phenolics Content and others, which were of very limited research value. Regarding the new method of weight gain, was it accurate enough to evaluate lipid oxidation? Did the authors consider the weight loss during oxidation, due to the release of some volatile compounds?
Author Response
The authors are grateful to the reviewer for his/her constructive evaluation of the manuscript entitled “A new insight into the weight gain method to monitor and evaluate lipid peroxidation” with the reference number foods-3430154.
Reviewer's comments and responses
- This study tried to test a new method based on the weight gain of oil upon oxidation. However, the study focus a lot on the characterization of the chosen three oils, including fatty acid composition, PV values, CDV, CV, AV, Total Tocopherols Content, Total Phenolics Content and others, which were of very limited research value.
Kindly, the oxidative stability of lipid systems is generally affected by their (1) fatty acid composition, (2) initial quality indicators (e.g. PV, AV, CD, and CV), and existing synthetic (e.g. BHA, BHT, and TBHQ) and/or natural antioxidants (e.g. tocopherols and phenolic compounds). As can be seen in different points of the manuscript, this compositional information helped us very well to interpret the oxidizabilities in terms of the calculated kinetic parameters. So, they were quite helpful and their being in the manuscript do not hurt the scientific quality of the paper.
- Regarding the new method of weight gain, was it accurate enough to evaluate lipid oxidation?
As stated in the sections R&D and Conclusions, the kinetic parameter Oiw (weight gain method) would be able to simulate well the kinetic parameter Oi (PV method), and the kinetic parameter rnw (for LOOH formation and polymerization) can provide some extra information in addition to those resulting from the parameter rn (for LOOH formation and decomposition). Anyway, as stated in the Conclusions section, study on a wider range of edible fats and oils under various antioxidative and pro-oxidative conditions would provide us with more certainty on the practical applications of the weight gain method.
- Did the authors consider the weight loss during oxidation, due to the release of some volatile compounds?
The present study deals with the easy and inexpensive measurement of change in weight and its synchronization with the change in PV. So, there was no direct measurement of the weight loss due to the release of volatile compounds.
Reviewer 3 Report
Comments and Suggestions for Authors Translator
The paper presents experimental data obtained as a result of monitoring the kinetics of change in peroxide value, PV, and weight gain during oxidation of canola, corn and peanut oil at 90 °C. In addition, it is worth noting that the authors have in-depth knowledge in the field of kinetics and mechanism of lipid oxidation.
I have some basic remarks and/or questions to the authors:
Why authors did not plan a purification of the studied vegetable oils, using, for example adsorption chromatography, to remove the minor components? Thus, the study of the uninhibited lipid autoxidation process can be a basis for the monitoring the inhibited (in the presence of tocopherols or other antioxidants and/or synergists) oxidation process.
Why are the graphs presented in Figures 1 and 2 in the Materials and Methods section and are they presented for the first time in the present study?
Author Response
The authors are grateful to the reviewer for his/her constructive evaluation of the manuscript entitled “A new insight into the weight gain method to monitor and evaluate lipid peroxidation” with the reference number foods-3430154.
Reviewer's comments and responses
The paper presents experimental data obtained as a result of monitoring the kinetics of change in peroxide value, PV, and weight gain during oxidation of canola, corn and peanut oil at 90 °C. In addition, it is worth noting that the authors have in-depth knowledge in the field of kinetics and mechanism of lipid oxidation.
- Why authors did not plan a purification of the studied vegetable oils, using, for example adsorption chromatography, to remove the minor components? Thus, the study of the uninhibited lipid autoxidation process can be a basis for the monitoring the inhibited (in the presence of tocopherols or other antioxidants and/or synergists) oxidation process.
There is no doubt that the kinetics of change in lipid peroxidation and weight gain would be affected by the minor components in edible fats and oils. In other words, all the kinetic parameters for a purified oil (by adsorption chromatography) would definitely be different from those for an oil inhibited by indigenous antioxidants/pro-oxidants. However, regardless of the chemical composition of edible fats and oils (canola, corn, and peanut in this study), the present study aimed to reveal whether the kinetics of change in weight and PV would be able to give similar relative results. This would provide us with a more comprehensive perspective regarding the use of the weight gain method in future studies.
- Why are the graphs presented in Figures 1 and 2 in the Materials and Methods section and are they presented for the first time in the present study?
Yes, they are presented for the first time, and they were relocated.
Reviewer 4 Report
Comments and Suggestions for Authors
The subject of the study is interesting, as it provides new kinetic insights in weight gain method used to evaluate oxidative stability of oils, which is of great practical importance.
The manuscript is well-written, with methods and results clearly demonstrated and thoroughly discussed.
Minor comment:
· The sentence in lines 34-37 “Such a usual peroxidation pathway, being composed…” should be rewritten more clearly. What is the difference between successive mechanism and stepwise mechanism?
· I recommend expressing concentration as molarity (M) rather than normality (N) – line 73.
· Line 75: please check instrument gas-liquid chromatograph, and add the model number.
· The quantities v/v , w/w as all physical quantities should be written in italics.
· Line 98 : The oils (0.04 – 1.00 g) – both values should have the same level of precision
· Equation 1: I recommend writing PV=kIP*t +PV0 instead PV=kIP(t)+PV0. The same comment applies to Equations 5 and 6.
· In line 164: rIP is zero-order rate constant, not the rate.
· I recommend adding a few more updated references if applicable.
Author Response
The authors are grateful to the reviewer for his/her constructive evaluation of the manuscript entitled “A new insight into the weight gain method to monitor and evaluate lipid peroxidation” with the reference number foods-3430154.
Reviewer's comments and responses
The subject of the study is interesting, as it provides new kinetic insights in weight gain method used to evaluate oxidative stability of oils, which is of great practical importance. The manuscript is well-written, with methods and results clearly demonstrated and thoroughly discussed.
Minor comment:
- The sentence in lines 34-37 “Such a usual peroxidation pathway, being composed…” should be rewritten more clearly. What is the difference between successive mechanism and stepwise mechanism?
The sentence was rewritten. The stepwise peroxidation, which its kinetic studies have recently been published (refs. 4 and 5) in detail, has clearly been shown in Figure 1 for the canola oil. Also, the corresponding kinetics have been provided in Tables 2 and 3.
- Jooyandeh, M.; Jaldani, S.; Farhoosh, R. Stepwise peroxidation of canola and olive oils: A kinetic study. J. Am. Oil Chem. Soc. 2023, 100, 975–983.
- Sarfaraz Khabbaz, E.; Jaldani, S.; Farhoosh, R. Unusual multiphase peroxidation of sunflower oil: a kinetic study. LWT-Food Sci. Technol. 2023, 184, 114981.
- I recommend expressing concentration as molarity (M) rather than normality (N) – line 73.
N was changed to M.
- Line 75: please check instrument gas-liquid chromatograph, and add the model number.
It was checked and the model was added.
- The quantities v/v, w/w as all physical quantities should be written in italics.
They were italicized.
- Line 98: The oils (0.04 – 1.00 g) – both values should have the same level of precision
It was corrected.
- Equation 1: I recommend writing PV=kIP*t +PV0 instead PV=kIP(t)+PV0. The same comment applies to Equations 5 and 6.
They were corrected.
- In line 164: rIP is zero-order rate constant, not the rate.
It was corrected.
- I recommend adding a few more updated references if applicable.
The latest references on the kinetics of lipid peroxidation, which have recently been published, were cited in the present study. As for the weight gain method, the references cited are relatively old but they are the latest ones of course.
Round 2
Reviewer 1 Report
Comments and Suggestions for Authors
The manuscript has been improved a lot after revision.
Comments on the Quality of English Languagenone
Author Response
The authors are grateful to the reviewer for his/her constructive evaluation of the manuscript.
Reviewer 2 Report
Comments and Suggestions for Authors
the authors did not address the raised questions properly, and the study had very limited research value
Comments on the Quality of English Languagethe language is ok
Author Response
In the first round, the respected reviewer raised three general questions that we did our best to respond them. The questions were not so challenging regarding the novel findings of work.